# Momentum Adversarial Distillation: Handling Large Distribution Shifts in Data-Free Knowledge Distillation

**Kien Do, Hung Le, Dung Nguyen, Dang Nguyen, Haripriya Harikumar,**
**Truyen Tran, Santu Rana, Svetha Venkatesh**
Applied Artificial Intelligence Institute (A2I2), Deakin University, Australia
*{k.do, thai.le, dung.nguyen, d.nguyen, h.harikumar,*
*truyen.tran, santu.rana, svetha.venkatesh}@deakin.edu.au*

## Abstract

Data-free Knowledge Distillation (DFKD) has attracted attention recently thanks to its appealing capability of transferring knowledge from a teacher network to a student network without using training data. The main idea is to use a generator to synthesize data for training the student. As the generator gets updated, the distribution of synthetic data will change. Such distribution shift could be large if the generator and the student are trained adversarially, causing the student to forget the knowledge it acquired at previous steps. To alleviate this problem, we propose a simple yet effective method called Momentum Adversarial Distillation (MAD) which maintains an exponential moving average (EMA) copy of the generator and uses synthetic samples from both the generator and the EMA generator to train the student. Since the EMA generator can be considered as an ensemble of the generator's old versions and often undergoes a smaller change in updates compared to the generator, training on its synthetic samples can help the student recall the past knowledge and prevent the student from adapting too quickly to new updates of the generator. Our experiments on six benchmark datasets including big datasets like ImageNet and Places365 demonstrate the superior performance of MAD over competing methods for handling the large distribution shift problem. Our method also compares favorably to existing DFKD methods and even achieves state-of-the-art results in some cases.

## 1 Introduction

With the development of deep learning, more pretrained deep neural networks have been released to the public [5, 10, 18, 38, 43]. However, their superior performances often come with big sizes, causing difficulties in deployment of these pretrained networks on resource-constrained devices. This leads to the demand for transferring knowledge from a cumbersome pretrained source network (called "teacher") to a compact target network (called "student") with a minimal loss of performance. This task is regarded as Knowledge Distillation (KD) [19].

The original idea of KD is to make use of class probabilities predicted by the teacher which encapsulate the hidden correlation among classes as training signals for the student. Note that such "dark" knowledge [19] is generally not available if the student is trained directly on raw data with one-hot labels. Later, various KD methods have been proposed to improve the quality of knowledge transfer. For example, AT [47] matches the spatial attention maps at intermediate layers of the student and the teacher. SPD [42] encourages the similarity between the student's and teacher's feature correlation matrices. PKT [36] transfers the conditional probability between two samples computed via kernel density estimation on the feature space. RKD [35] exploits relational knowledge for distillation. VID

36th Conference on Neural Information Processing Systems (NeurIPS 2022).

[1] maximizes a variational lower bound of the mutual information (MI) between the student's and teacher's representations. CRD [41] uses contrastive learning as a proxy for maximizing MI.

The common drawback of these methods is the reliance on samples from the teacher training set. However, in practice, accessing the original training data is usually infeasible due to many reasons such as data privacy (e.g., healthcare data containing personal information) or data regarded as intellectual property of the vendors. Addressing this critical issue, Data-Free Knowledge Distillation (DFKD) methods have been introduced [7, 13, 27, 29, 31, 34, 44–46]. A common DFKD approach is to use a generator network to synthesize training data and jointly train the generator and the student in an adversarial manner [13, 31, 46]. Under this adversarial learning scheme, the student attempts to make predictions as close as possible to the teacher's on synthetic data generated by the generator, while the generator tries to create samples that maximize the mismatch between the student's and the teacher's predictions. This adversarial game enables a rapid exploration of synthetic distributions useful for knowledge transfer between the teacher and the student. At the same time, it could also lead to large shifts in the synthetic distributions, causing the student to forget useful knowledge acquired at the previous steps and suffer from performance drops [3].

In this paper, we propose a simple yet effective method called Momentum Adversarial Distillation (MAD) to mitigate the large distribution shift problem in adversarial DFKD. MAD maintains an exponential moving average (EMA) copy of the generator which is responsible for storing information about past updates of the generator. By using synthetic samples from the EMA generator as additional training data for the student besides those from the generator, MAD can ensure that the student can recall the old knowledge, hence, is less prone to forgetting. Moreover, to reduce the negative effect caused by spurious solutions of an unconditional generator when learning on large datasets such as ImageNet, we propose to use a class-conditional generator that takes the sum of a noise vector and a class embedding vector as input, and train this generator with a new objective that suppresses the presence of spurious solutions. This technique requires only a small change in the generator's architecture but enables MAD (and possibly other adversarial DFKD methods) to learn surprisingly well on large datasets. Through extensive experiments on three small and three large image datasets, we demonstrate that our proposed method is far better than related baselines [3, 31] in dealing with the large distribution shift problem. In some cases, MAD even outperforms current state-of-the-art methods [8, 14].

## 2 Adversarial Data-Free Knowledge Distillation

Let $\mathtt{T}$ be a *teacher* network pretrained on some dataset $\mathcal{D}_{\text{train}}$ and $\mathtt{S}$ be a fresh *student* network. Let $\mathtt{T}(\cdot)$ and $\mathtt{S}(\cdot)$ denote outputs of the teacher and student networks *before the softmax activation*, respectively. In Data-Free Knowledge Distillation (DFKD), we want to transfer knowledge from $\mathtt{T}$ to $\mathtt{S}$ so that $\mathtt{S}$ performs as well as or even better than $\mathtt{T}$ on the original test set $\mathcal{D}_{\text{test}}$ but with a constraint that no training data for $\mathtt{S}$ is available. An intuitive way to deal with such constraint is learning an additional generator network $\mathtt{G}$ that can generate synthetic data for training $\mathtt{S}$ from a noise distribution $p(z)$ usually chosen to be the standard Gaussian distribution $\mathcal{N}(0, \mathrm{I})$. Adversarial Belief Matching (ABM) [31] proposed an adversarial learning framework between $\mathtt{S}$ and $\mathtt{G}$ via optimizing the following min-max objective:

$$\min_{\mathtt{S}} \max_{\mathtt{G}} \mathbb{E}_{z \sim p(z)} \left[ \mathcal{L}_{\text{KD}}(\mathtt{G}(z)) \right] \tag{1}$$

$$\Leftrightarrow \min_{\mathtt{S}} \max_{\mathtt{G}} \mathbb{E}_{z \sim p(z), x = \mathtt{G}(z)} \left[ \mathcal{L}_{\text{KD}}(x) \right], \tag{2}$$

where $\mathcal{L}_{\text{KD}}(x)$ denotes the knowledge distillation (KD) loss, i.e., the discrepancy between $\mathtt{S}(x)$ and $\mathtt{T}(x)$. In ABM, $\mathcal{L}_{\text{KD}}(x)$ is the Kullback-Leibler (KL) divergence between class probabilities of $\mathtt{T}$ and $\mathtt{S}$ computed on $x$:

$$\mathcal{L}_{\text{KD}}(x) \triangleq D_{\text{KL}} \left( \mathtt{Tp}(x) \| \mathtt{Sp}(x) \right) = \sum_{c=1}^{C} \mathtt{Tp}(x)[c] \cdot \left( \log \mathtt{Tp}(x)[c] - \log \mathtt{Sp}(x)[c] \right), \tag{3}$$

where $\mathtt{Tp}(x) = \text{softmax}(\mathtt{T}(x))$ and $\mathtt{Sp}(x) = \text{softmax}(\mathtt{S}(x))$ denote the class probabilities of $\mathtt{T}$ and $\mathtt{S}$ computed on $x$, respectively; and $C$ is the total number of classes.

The core idea behind the optimization in Eq. 1 is to encourage $\mathtt{G}$ to generate samples on which the outputs of $\mathtt{S}$ are very different from those of $\mathtt{T}$ (or the KD loss is large). Typically, the generated

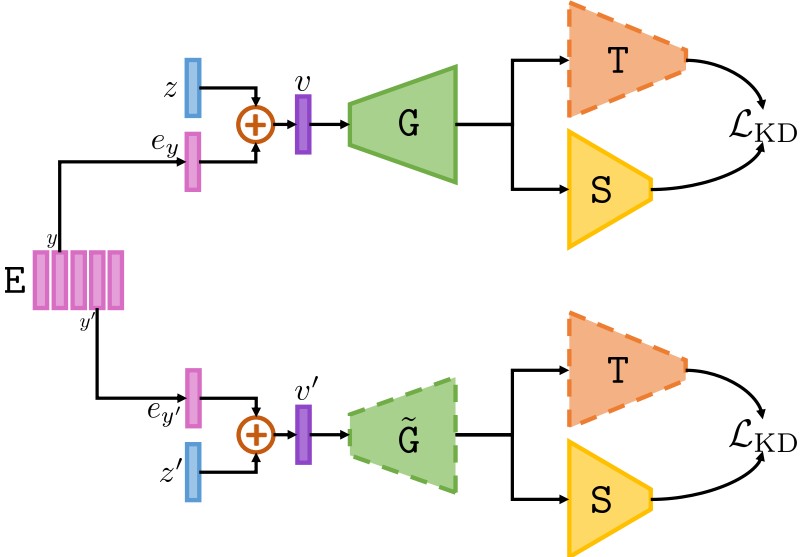

Figure 1: An illustration of our proposed *Momentum Adversarial Distillation (MAD)* consisting of a teacher (T), a student (S), a class-conditional generator (G), an EMA generator (G̃), and class embeddings (E). Networks with dashed borders (T, G̃) are not optimized during training. $z$ and $z'$ are random noises sampled from $\mathcal{N}(0, \mathrm{I})$. $e_y$ and $e_{y'}$ are gathered from E at index $y$ and $y'$, respectively.

samples have not been observed by S during training (otherwise, the KD loss will be small) and the learning task of S is to match T on these novel samples. It is expected that in the ideal case, via continuous adversarial exploration of G, S can be exposed to a diverse enough set of synthetic samples which allows S to match T with an arbitrarily small prediction error on $\mathcal{D}_{\text{test}}$.

In practice, we usually implement Eq. 1 by alternately optimizing S and G in $n_{\mathtt{S}}$ and $n_{\mathtt{G}}$ steps, respectively. Thus, to avoid confusion later, we refer to a $n_{\mathtt{S}}$-step update of S as an update *stage* of S and similarly, a $n_{\mathtt{G}}$-step update of G as an update *stage* of G.

## 3 Momentum Adversarial Distillation

### 3.1 Handling large distribution shifts with an additional EMA generator

Although theoretically sound, the optimization in Eq. 1 has a practical problem: If the update of G varies too much, the distribution of synthetic samples generated by G will change significantly over two consecutive update steps of S, which in turns causes S to catastrophically forget what it has learned at the previous stages [3] to adapt to the new update of G. In order to address this problem, we propose to maintain an exponential moving average (EMA) of the generator G, denoted by G̃, during learning and use synthetic samples from both G and G̃ to train S as follows:

$$\min_{\mathtt{S}} \mathcal{L}_{\mathtt{S}} \triangleq \lambda_0 \mathbb{E}_{z \sim p(z)} \left[ \mathcal{L}_{\text{KD}}(\mathtt{G}(z)) \right] + \lambda_1 \mathbb{E}_{z' \sim p(z)} \left[ \mathcal{L}_{\text{KD}}(\tilde{\mathtt{G}}(z')) \right], \tag{4}$$

where $\lambda_0, \lambda_1 \geq 0$ are coefficients. The parameters $\theta_{\tilde{\mathtt{G}}}^t$ of the momentum generator G̃ at the update *stage* $t$ of S (after $n_{\mathtt{G}}$ steps update of G) are computed as:

$$\theta_{\tilde{\mathtt{G}}}^t = \alpha \cdot \theta_{\tilde{\mathtt{G}}}^{t-1} + (1 - \alpha) \cdot \theta_{\mathtt{G}}^t, \tag{5}$$

where $\alpha$ ($0 < \alpha < 1$) is the momentum. If $\alpha$ is close to 1, G̃ will change very slightly compared to G. Therefore, by using synthetic samples from G̃ as additional training data for S besides those from G, we can alleviate the large exploratory distribution shift caused by the large update of G and achieve a stable update of S. We name our proposed method *Momentum Adversarial Distillation* (MAD). See Fig. 1 for an illustration.

## 3.2 Enabling MAD to learn well on large datasets

During our experiments, we observed that MAD, with the unconditional generator G described in Section 2, is not able to learn well on large datasets such as ImageNet. Our hypothesis is that in case of large datasets, the objective of G in Eq. 2 induces a very large number spurious solutions, which hampers the learning of G. To see this, first let us recall the objective of G which is maximizing the KL divergence between $\mathrm{Sp}(x)$ and $\mathrm{Tp}(x)$ over some synthetic sample $x = \mathrm{G}(z)$. If we, for example, assume that there are 3 classes in total and $\mathrm{Tp}(x)$ is fixed at $[1, 0, 0]$, then we could train G to generate $x$ so that $\mathrm{Sp}(x)$ is either $[0, 1, 0]$ or $[0, 0, 1]$. In this toy example, we see 2 spurious solutions for 3 classes. If a dataset has 1000 classes like ImageNet, there will be at least 999 spurious solutions. Besides class numbers, larger input sizes also increase the space and the number of (spurious) solutions. Unfortunately, the more spurious solutions, the more likely G could jump from one spurious solution to another in successive update steps, causing instability in training G.

To overcome this problem, we propose to condition G on a class label $y$ as $\mathrm{G}(z + e_y)$ where $e_y$ is a trainable embedding of $y$. We found that using the sum of $z$ and $e_y$ as input to G instead of concatenation allows our method to learn much better, possibly because the noise in updating $e_y$ is absorbed into the stochasticity of $z$ via summation rather than concatenation. We illustrate this idea in Fig. 1, and provide an empirical justification in Appdx. B.3. Denoted by E the list of trainable embedding vectors for all classes ($\mathrm{E} = (e_1, ..., e_C)$), we train both G and E together by minimizing the following loss:

$$\min_{\mathrm{G,E}} \mathcal{L}_{\mathrm{G,E}} \triangleq \mathbb{E}_{z \sim \mathcal{N}(0,\mathrm{I}), y \sim \mathrm{Cat}(C), x=\mathrm{G}(z+e_y)} \left[ -\lambda_2 \mathcal{L}_{\mathrm{KD}}(x) + \lambda_3 \mathcal{L}_{\mathrm{NLL}}(x, y) + \lambda_4 \mathcal{L}_{\mathrm{NormReg}}(e_y) \right], \quad (6)$$

where $\mathrm{Cat}(C)$ is the uniform categorical distribution over $C$ classes; $\lambda_2, \lambda_3, \lambda_4 \geq 0$ are hyperparameters; and $\mathcal{L}_{\mathrm{NLL}}(x, y)$ and $\mathcal{L}_{\mathrm{NormReg}}(e_y)$ are the *negative log-likelihood* and the *norm regularization* losses respectively, defined as follows:

$$\mathcal{L}_{\mathrm{NLL}}(x, y) \triangleq -\log \mathrm{Tp}(x)[y], \quad (7)$$

$$\mathcal{L}_{\mathrm{NormReg}}(e_y) \triangleq \max \left( \|e_y\|_2 - \gamma \times \sqrt{d_e}, 0 \right), \quad (8)$$

where $d_e$ denotes the dimensionality of $e_y$; and $\gamma \geq 1$ is a scaling hyperparameter. $\mathcal{L}_{\mathrm{NormReg}}(e_y)$ servers as a constraint that restricts the norm of $e_y$ to be smaller than $\gamma \times \sqrt{d_e}$. An explanation of the formula $\gamma \times \sqrt{d_e}$ is provided in Appdx. C. Intuitively, via minimizing $\mathcal{L}_{\mathrm{NLL}}(x, y)$, G could maintain its focus on predicting $y$ throughout its entire update stage rather than jumping between different spurious solutions. Besides, since $y$ is sampled uniformly, the synthetic data will be evenly distributed among classes.

In case the teacher T has BatchNorm [20] layers, we can make use of the running mean $\bar{\mu}_\ell$ and running variance $\bar{\omega}_\ell$ of each BatchNorm layer $\ell$ of T to guide the data synthesis of G by adding the BatchNorm moment matching (BNmm) loss [45] below to $\mathcal{L}_{\mathrm{G,E}}$ (weighted by $\lambda_5 \geq 0$):

$$\mathcal{L}_{\mathrm{BNmm}} \triangleq \sum_\ell \|\mu_\ell - \bar{\mu}_\ell\|_2^2 + \|\omega_\ell - \bar{\omega}_\ell\|_2^2, \quad (9)$$

where $\mu_\ell, \omega_\ell$ are the empirical mean and variance of the features at the BatchNorm layer $\ell$ w.r.t. the batch of synthetic samples from G. During our experiment, we observed that using $\mathcal{L}_{\mathrm{BNmm}}$ improves knowledge distillation of the student S.

## 4 Related Work

In Data-Free Knowledge Distillation (DFKD), it is critical to synthesize data that are useful for transferring knowledge from T to S. Existing DFKD methods differ mainly in their objectives to guide the data synthesis. These methods generally fall into either the adversarial camp or the non-adversarial camp. Non-adversarial DFKD methods [7, 27, 29, 34, 44, 46] make use of certain heuristics to search for synthetic data that resembles the original training data $\mathcal{D}_{\mathrm{T,train}}$. For example, ZSKD [34] and DAFL [7] consider prediction probabilities of T and the confidence of T as heuristics. In DAFL, synthetic data is generated via a generator G rather than being optimized directly like in ZSKD. KegNet [46] extends DAFL by making G conditioned on class labels. Adversarial DFKD methods [8, 14, 17, 37, 45, 50] leverage adversarial learning to explore the data space more efficiently.

Most methods of this type are derived from the ABM [31] discussed in Section 2 with additional objectives to improve the quality and/or diversity of synthetic data. For example, RDSKD [17] uses a diversity-seeking loss [30]; CMI [14] uses an inverse contrastive loss to improve diversity; and DeepInversion [45] uses Batch Norm moment matching (BNmm) and DeepDream's inception losses (total variation, L2) [32] to generate visually interpretable images.

DeepInversion [45] is one of a few DFKD methods [12, 29, 45] that have been shown to learn well on the full-size ImageNet dataset. Its success has inspired subsequent works on large-scale continual learning [39] and data-free object detection [6]. However, training DeepInversion on ImageNet is very time-consuming because this method optimizes synthetic images directly and *each* complete optimization requires a very large number of iterations (about 20,000). LS-GDFD [29] learns to generate synthetic data instead but requires one generator for each class (that is, 1,000 generators for 1,000 classes) to avoid the "mode collapse" problem and achieve good performance. FastDFKD [12] leverages meta-learning to speed up the training process significantly. On the contrary, by applying our proposed technique in Section 3.2, our MAD only needs 1 generator (compared to 1,000 of LS-GDFD) and 6,000 training iterations (compared to 20,000× of DeepInversion) to achieve reasonably good performance on ImageNet (Section 5.2.2). In addition, our technique is orthogonal to and can combine with the meta-learning idea in [12] for further improvement.

Besides the DFKD methods discussed above, there is a line of works that consider a less extreme scenario in which unlabeled transfer sets are given for training. These transfer sets can be very different from the original dataset in terms of distribution and semantics and can be freely collected from open resources. Nayak et al. [33] analyzed various kinds of transfer sets ranging from random noise images to natural images and found that the target-class balance property of the transfer set and the similarity between the transfer set and the original training set play important roles in improving knowledge distillation results. Fang et al. [11] argued that although the transfer images can be semantically different from the original traning images, they still share common local visual patterns. Therefore, the authors proposed an interesting method called MosaicKD which combines different local patches extracted from the transfer images to craft synthetic mosaic images that capture the semantics of the original data while enjoy realistic local structures.

Adversarial training methods like GANs has been shown to suffer from the catastrophic forgetting problem of the discriminator [26, 40]. These works consider GANs training as a continual learning problem and leverage existing methods to mitigate the issue. For example, Liang et al. [26] use either EWC [23] or SI [49] to enforce similarity between the discriminator's weights at the current step and at the last checkpoint. Hoang et al. [40] suggest some other tricks such as using a momentum optimizer (e.g., Adam [21]) or applying gradient penalty [16] to the discriminator. In this paper, we consider the catastrophic forgetting problem of the student as a consequence of the large distribution shift problem caused by the generator. Therefore, we focus on regularizing the generator rather than the student. Besides, from this perspective, the catastrophic forgetting problem in GANs is indeed less severe than that in adversarial DFKD. It is because in GANs, the discriminator is trained on real data with a fixed distribution, which can somewhat reduce the effect of the distribution shift of fake (synthetic) data while in adversarial DFKD, real data is not available. Recent works that also address the student forgetting problem in DFKD like ours include DFKD-Mem [3] and PRE-DFKD [2]. DFKD-Mem [3] stores the past synthetic samples in a memory bank and uses these samples as additional training data for the student. Our method, on the other hand, uses an EMA generator to generate old synthetic samples on-the-fly, which is more memory efficient and adapts better to the student update. PRE-DFKD [2] models past synthetic data via a VAE [22] and treats the decoder of this VAE as a replay generator. However, training a VAE on a continuous stream of synthetic samples could be unstable and could lead to another catastrophic forgetting problem on its own. In addition, VAE is not effective for generating images with large size (e.g., ImageNet images) due to the "posterior collapse" problem [28].

## 5 Experiments

### 5.1 Experimental Setup

**Datasets** We consider the image classification task and evaluate our proposed method on 3 small image datasets (CIFAR10 [24], CIFAR100 [24], TinyImageNet [25]), and 3 large image datasets (ImageNet [9], Places365 [51], Food101 [4]). Details are provided in Appdx. A.1.

| Dataset | Arch. | Tea.$^a$ | Stu.$^a$ | DAFL$^a$ | ZSKT$^a$ | ADI$^a$ | DFAD$^b$ | DDAD$^b$ | DFQ$^a$ | CMI$^a$ | MAD |
|---------|-------|------|------|------|------|------|------|------|------|------|------|
| CIFAR10 | ♡ | 95.70 | 95.20 | 92.22 | 93.32 | 93.26 | 93.30 | 94.81 | 94.61 | *94.84* | **94.90** |
|  | ♢ | 94.87 | 93.95 | 81.55 | 89.66 | 89.72 | - | - | 92.01 | *92.52* | **92.64** |
| CIFAR100 | ♡ | 78.05 | 77.10 | 74.47 | 67.74 | 61.32 | 69.43 | 75.04 | 77.01 | 77.04 | **77.31** |
|  | ♢ | 75.83 | 73.56 | 40.00 | 28.44 | 61.34 | - | - | 59.01 | **68.75** | *64.05* |
| TinyIN | ♡ | 66.44 | 64.87 | - | - | - | - | - | *63.73* | **64.01** | 62.32 |

Table 1: Classification accuracy (in %) of the student trained by different DFKD methods on 3 small image datasets. The teacher/student architecture settings are ResNet34/ResNet18 (♡) and WRN40-2/WRN16-2 (♢). $^a$ and $^b$ denote results taken from [14] and [50], respectively. Tea. and Stu. denote the teacher and student trained from scratch on $\mathcal{D}_{\text{train}}$. The best and second best results are highlighted in **bold** and *italic*, respectively.

**Network architectures**   For the small datasets, we follow [8, 14] and consider ResNet34/ResNet18 [18] and WRN40-2/WRN16-2 [48] for the teacher/student. We observed that knowledge distillation with the WRN architectures is more challenging than with the ResNet architectures, possibly because WRN40-2/WRN16-2 have much fewer parameters than ResNet34/ResNet18. For the large datasets, we use the AlexNet architecture for both the teacher and student for fast training. Since AlexNet does not have any BatchNorm layer, we exclude the BNmm loss (Eq. 9) from the total loss of the generator in our experiments on the large datasets. Architectures of the generator w.r.t. different datasets are given in Appdx. A.4.

**Training settings of teacher**   For ImageNet, we make use of pretrained networks provided by PyTorch. For other datasets, we train the teacher from scratch. Detailed training settings of the teacher are given in Appdx. A.2.

**Training settings of MAD**   If not otherwise specified, we set the momentum $\alpha$ in Eq. 5 to 0.95 and the length of the noise vector to 256. We train the student S using SGD and Adam for the small and large datasets, respectively. We train the generator G using Adam for both the small and large datasets. To reduce the difficulty of training G for the large datasets, we pretrain G for some steps before the main KD training by setting the coefficient of $\mathcal{L}_{\text{KD}}$ to 0 and only optimizing the remaining losses in Eq. 6. The generator G is class-conditional (as described in Section 3.2) for the large datasets, and unconditional for the small datasets. We empirically found that this leads to better results. For further details about the training settings of MAD, please refer to Appdx. A.3.

## 5.2   Results

### 5.2.1   Comparison with existing DFKD methods

In Table 1, we compare MAD with existing DFKD methods [7, 8, 13, 14, 34, 45, 50] on the small datasets. The results of baselines are taken from [14, 50]. We checked the results of the teacher trained by us and found that they are quite similar to the results of the teacher reported in [14] and in Table 1 (details in Appdx. B.1). This means our results of MAD are comparable with those of the baselines. We see that MAD outperforms all the DFKD methods on CIFAR10 and CIFAR100 in case the teacher and student are ResNets [18] (♡). In case the teacher and student are WideResNets [48] (♢), our method still achieves significantly better results than most of the baselines such as DAFL [7] and ZSKT [34] and only performs worse than CMI [14]. We hypothesize it is mainly because of the differences in model design and training settings of CMI and MAD. For example, CMI has an additional contrastive-loss-based objective that encourages the generator's diversity while MAD does not. CMI also performs more updates of the student and generator per training step than our method[1].

However, our ultimate goal for this section is *not* to show that MAD can achieve state-of-the-art results in all cases (which often requires intensive hyper-parameter tuning) but to verify the soundness of our implementation of MAD. In order to see clearly the advantage of MAD, we need to compare MAD with its related baselines *under the same settings*. This will be presented in Section 5.2.2.

---

[1]We could not find the supplementary material containing the training settings of CMI. However, from the official code provided by the authors, we saw that they set $n_{\text{G}} = 200$ and $n_{\text{S}} = 2000$ for WideResNet teacher/student on CIFAR10 while we set $n_{\text{G}} = 3$ and $n_{\text{S}} = 60$ (see Appdx. A.3). In general, setting larger $n_{\text{G}}$ and $n_{\text{S}}$ often leads to better results (Section 5.3) but will increase the training time.

| Dataset | CIFAR10 | CIFAR100 | TinyIN | ImageNet | Places365 | Food101 |
| Arch. | ◇ | ◇ | ♡ | ♣ | ♣ | ♣ |
|---|---|---|---|---|---|---|
| Teacher | 94.65 | 75.65 | 66.47 | 56.52 | 50.80 | 65.15 |
| ABM | 92.38 | 62.59 | 59.75 | 41.23 | 41.84 | 60.37 |
| DFKD-Mem | 92.09 | 61.25 | 58.66 | 43.30 | 42.38 | 61.25 |
| MAD | **92.64** | **64.05** | **62.32** | **45.48** | **43.67** | **61.74** |

Table 2: Classification accuracy (in %) of MAD and its related baselines on 3 small and 3 large image datasets. The teacher/student architecture settings are ResNet34/ResNet18 (♡), WRN40-2/WRN16-2 (◇), and AlexNet/AlexNet (♣). The teacher's results are from our own runs (Appdx. B.1). The best results are highlighted in **bold**.

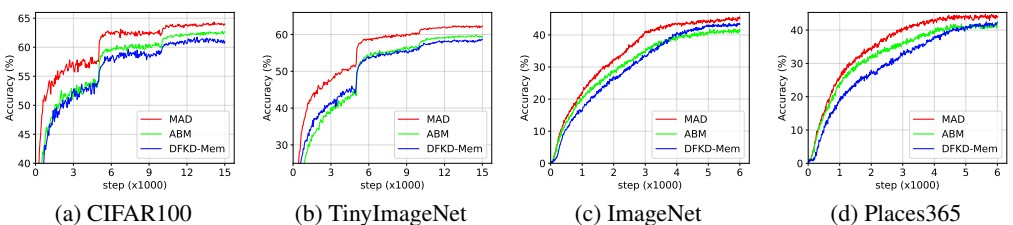

|   (a) CIFAR100   |   (b) TinyImageNet   |   (c) ImageNet   |   (d) Places365   |

Figure 2: Test accuracy curves of S trained via MAD, ABM, and DFKD-Mem on some datasets.

#### 5.2.2 Comparison with related baselines

We consider two related baselines of MAD which are ABM [31] and DFKD-Mem [3]. ABM learns the student S with only synthetic samples from G. DFKD-Mem, on the other hand, stores past synthetic samples in a memory bank, and uses samples from this memory bank (dubbed "memory samples") as well as those generated by G as training data for S. We trained ABM and DFKD-Mem using exactly the same settings for training MAD. For DFKD-Mem, we set the memory size to 8,192. For other memory sizes, the results remain relatively similar as shown in Appdx. B.2.

From Table 2 and Fig. 2, it is clear that MAD significantly outperforms both ABM and DFKD-Mem on all datasets. In addition, the performance gaps between our method and the two baselines tend to be larger for larger datasets. For example, MAD achieves about 1.5/2.8%, 2.5/3.6%, and 4.2/2.2% higher accuracy than ABM/DFKD-Mem on CIFAR100, TinyImageNet, and ImageNet, respectively. These empirical results suggest the importance of the EMA generator $\tilde{G}$ in mitigating the large distribution shift caused by G.

In this experiment, we found that DFKD-Mem often performs worse than ABM on CIFAR100 and TinyImageNet. We found the decay of the student learning rate is the main reason for this. As shown in Fig. 3a, the distillation loss on memory samples surges when the (student) learning rate is decayed and cannot recover if the new learning rate is too small ($lr_S = 1e-4$), which is in contrast to the distillation loss on synthetic samples from G or $\tilde{G}$ (Figs. 3b,3c). This implies a potential issue of storing old samples in a memory bank instead of using the EMA generator as memory samples could be completely out-of-date if the student suddenly change its state (e.g., via learning rate decay). However, even when the learning rate does not change (e.g., from step 0 to step 100 on CIFAR100 or on ImageNet/Places365), DFKD-Mem still performs worse than our method.

#### 5.2.3 Comparing the changes in update of $\tilde{G}$ and G

In order to see whether $\tilde{G}$ actually has smaller changes in update than G or not, we perform the following experiment: Let $G_t$ and $\tilde{G}_t$ be the versions of the generator and the EMA generator respectively at step $t$, and $S_{t-\tau}$ be the version of the student at step $t - \tau$ ($0 < \tau < t$). We then measure two different average Jensen-Shannon (JS) divergences between the prediction probabilities of $S_{t-\tau}$ and T on two separate sets of synthetic samples from $G_t$ and $\tilde{G}_t$. We hypothesize that if G has a smaller change in the distribution of synthetic samples than G, S will memorize the samples from $\tilde{G}$ more and will match T better on those samples, which leads to a smaller average JS divergence. This hypothesis is clearly reflected on results in Fig. 4, which verifies the reasonability of using $\tilde{G}$ in alleviating the large distribution shift problem caused by G.

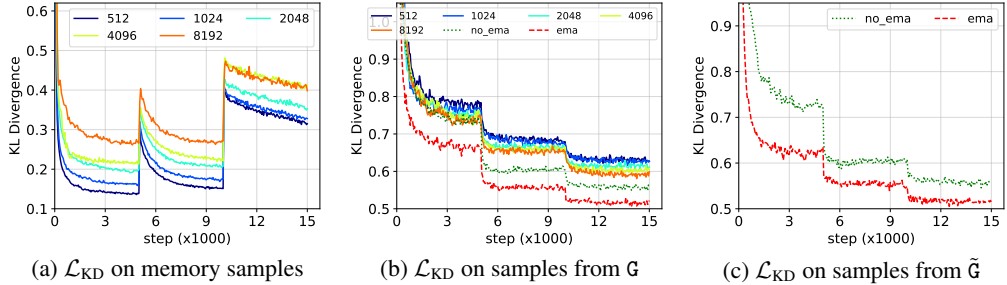

(a) $\mathcal{L}_{KD}$ on memory samples     (b) $\mathcal{L}_{KD}$ on samples from G     (c) $\mathcal{L}_{KD}$ on samples from $\tilde{G}$

Figure 3: Distillation loss curves on memory samples (a) and samples generated by G (b) and $\tilde{G}$ (c). The numbers in the legends denote DFKD-Mem with the corresponding memory sizes. "no_ema" and "ema" denote ABM and MAD, respectively.

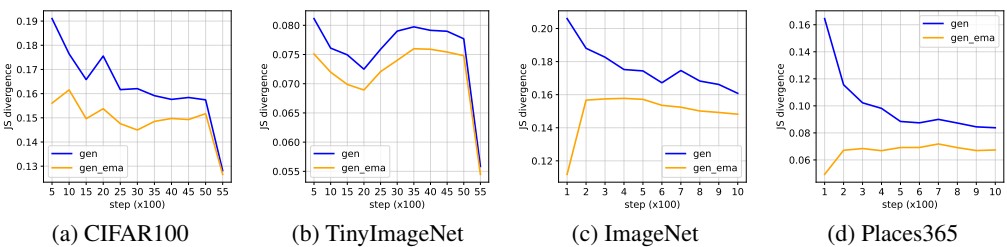

(a) CIFAR100     (b) TinyImageNet     (c) ImageNet     (d) Places365

Figure 4: Average Jensen-Shannon divergences between the prediction probabilities of $S_{t-\tau}$ and $T$ computed on synthetic samples from $G_t$ (labeled as "gen") and $\tilde{G}_t$ (labeled as "gen_ema") for different datasets. For CIFAR100 and TinyImageNet, we set $t \in [500, 5500]$ with step size of 500 and $\tau = 50$. For ImageNet and Places365, we set $t \in [100, 1000]$ with step size of 100 and $\tau = 10$. Note that the sudden drops at step 5000 in (a), (b) correspond to the decay of the learning rate by 0.1.

### 5.2.4 Visualization of synthetic samples

In Fig. 5, we visualize the synthetic data generated by G and $\tilde{G}$. Although the generated images are not visually realistic, they are visually diverse, suggesting no mode collapse has occurred during training MAD. Besides, samples generated by $\tilde{G}$ are different from those generated by G which indicates that the EMA generator could act as a complement for the generator in our model.

### 5.3 Sensitivity Analysis

Below we investigate some choices of hyperparameters that could affect the performance of MAD. Unless stated otherwise, the dataset we use is CIFAR100.

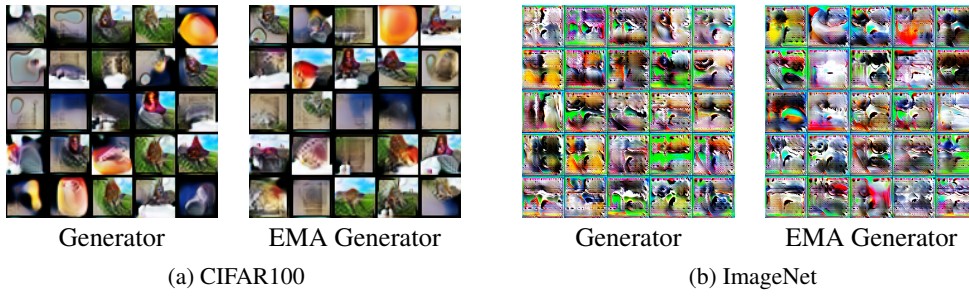

Generator     EMA Generator        Generator     EMA Generator

(a) CIFAR100           (b) ImageNet

Figure 5: Synthetic data generated by G and $\tilde{G}$ in case the original dataset is CIFAR100 (a) and ImageNet (b).

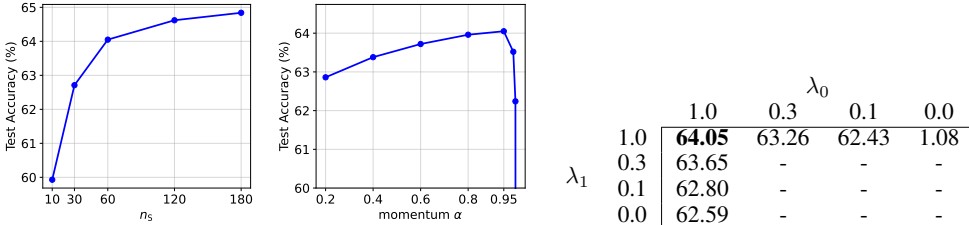

Figure 6: Test accuracy of our method w.r.t. different values of student update steps $n_{\mathtt{S}}$ (**left**), of the momentum $\alpha$ (**middle**), and of the coefficients $(\lambda_0, \lambda_1)$ in Eq. 4 (**right**). The dataset is CIFAR100.

**Different values of the coefficients in $\mathcal{L}_{\mathtt{S}}$**   We can control the relative importance of samples from $\tilde{\mathtt{G}}$ over those from $\mathtt{G}$ by changing the values of the two coefficients $\lambda_0$ and $\lambda_1$ in $\mathcal{L}_{\mathtt{S}}$ (Eq. 4). As shown in Fig. 6 (right), the best result is obtained when $\lambda_0 = \lambda_1 = 1$ which is our default setting for MAD. Decreasing either $\lambda_0$ or $\lambda_1$ will lead to worse performance. In the worst case when $\lambda_0 = 0$ and $\lambda_1 = 1$, the model learns for some epochs and then suddenly stops learning. The main reason is that $\tilde{\mathtt{G}}$ often updates much slower than $\mathtt{G}$ and $\mathtt{S}$ so we need adversarial samples from $\mathtt{G}$ to keep $\mathtt{S}$ learning. Otherwise, $\mathtt{S}$ will overfit the samples from $\tilde{\mathtt{G}}$ and learn nothing.

**Different values of the momentum $\alpha$**   In Fig. 6 (middle), we show the classification accuracy of the student of MAD with $\alpha$ in $\{0.2, 0.4, 0.6, 0.8, 0.95, 0.99, 0.999, 1.0\}$. We observe that the performance of our method increases when $\alpha$ becomes larger as $\tilde{\mathtt{G}}$ is more different from $\mathtt{G}$. However, if $\alpha$ is too large (e.g., 0.999), the performance drops since the update of $\mathtt{G}$ is very small, resulting in almost no change in the distribution of synthetic samples. In case $\alpha = 1.0$, MAD only achieves about 50% prediction accuracy on test data.

**Different update steps of $\mathtt{S}$**   From Fig. 6 (left), we see that increasing the number of update steps for $\mathtt{S}$ usually leads to better performances of MAD since $\mathtt{S}$ learns to match $\mathtt{T}$ better. In exchange, the training time will increase.

## 6   Conclusion

We have presented Momentum Adversarial Distillation (MAD), a simple yet effective method to deal with the large distribution shift problem in adversarial Data-Free Knowledge Distillation (DFKD). MAD maintains an exponential moving average (EMA) copy of the generator which, by design, encapsulates information about past updates of the generator and is updated at a slower pace than the generator. By training the student on samples from both the generator and the EMA generator, MAD can prevent the student from adapting too much to the generator at the current step and forgetting old knowledge learned from the generator at the previous steps. We have also described a new type of conditional generator along with a new loss for training it which enable our model to learn well on large datasets. Our experiments on various datasets demonstrated the superior performance of our method over competing baselines that either use only a generator or use a memory bank in place of an EMA generator.

We note that our idea of using an EMA generator to mitigate large distribution shifts is general and can be generalized to other machine learning problems besides DFKD. For example, the technique can be adapted for general continual learning (in which large distribution shifts can happen gradually or suddenly with no clear task boundaries), and source-data-free domain adaptation. Our method is also well suited for DFKD with other data types such as video, text, or graph.

**Limitations**   In our current implementation of MAD, we perform an additional forward pass through $\tilde{\mathtt{G}}$ for every training step of $\mathtt{S}$. This increases the total training time of by about 40% compared to ABM. However, this technical problem can be somewhat addressed by first storing synthetic samples from $\mathtt{G}$ and $\tilde{\mathtt{G}}$ in a buffer before each training stage of $\mathtt{S}$ and then training $\mathtt{S}$ with samples from the buffer only. This will be left for future work.

**Negative Social Impacts** The DFKD problem may have negative social impacts related to data privacy as generated data could somehow reveal the original training data. However, as shown in Fig. 5, our proposed method does not attempt to improve the visual interpretability of synthetic data but addresses the large distribution shift problem. This target seems to be harmless to the society.

## Acknowledgement

This research was partially funded by the Australian Government through the Australian Research Council (ARC). Prof. Venkatesh is the recipient of an ARC Australian Laureate Fellowship (FL170100006).

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
