# A Experimental Setup

## A.1 Datasets

In Table 3, we provide information about the image size, the number of classes, and the number of training/test samples of the datasets used in our experiment.

| Datatset | Image size | #classes | #train | #test |
|---|---|---|---|---|
| CIFAR10 | $3 \times 32 \times 32$ | 10 | 50,000 | 10,000 |
| CIFAR100 | | 100 | 50,000 | 10,000 |
| Tiny-ImageNet | $3 \times 64 \times 64$ | 200 | 100,000 | 10,000 |
| ImageNet | | 1000 | 1,281,167 | 50,000 |
| Places365 | $3 \times 256 \times 256$ | 365 | 1,803,460 | 36,500 |
| Food101 | | 101 | 75,750 | 25,250 |

Table 3: Details of the datasets used in our experiment.

## A.2 Training Settings of Teacher

We provide training settings of the teacher w.r.t. different datasets in Table 4.

| Dataset | Training settings | | | | | | | | | |
|---|---|---|---|---|---|---|---|---|---|---|
| | $opt$ | $lr$ | $wd$ | $mo$ | $bs$ | $ls$ | $ld$ | $ldep$ | $ep$ | $wep$ |
| CIFAR10/100 | SGD | 0.1 | 5e-4 | 0.9 | 128 | No | 0.1 | 80, 120 | 160 | 0 |
| TinyImageNet | | | | | 256 | No | | | | |
| Food101 | | 0.01 | 1e-4 | | 512 | Yes | | 30, 60 | 90 | 5 |
| Places365 | | | | | 1024 | Yes | | | | |

Table 4: Training settings of teacher w.r.t. different datasets. Meanings of abbreviations: $opt$: optimizer, $lr$: learning rate, $wd$: weight decay, $mo$: momentum, $bs$: batch size, $ls$: scaling learning rate or not with the base batch size of 256 [15], $ld$: learning rate decay, $ldep$: epochs at which learning rate are decayed, $ep$: total number of epochs, $wep$: number of warm-up epochs.

## A.3 Training Settings of MAD

In Tables 5,6,7, we provide the training settings of MAD used in this paper. Despite multiple attempts, we could not find a global configuration that works well for all datasets and architectures.

In practice, we do *not* optimize the student and the generator via the plain losses in Eq. 4 and Eq. 6, respectively but with some additional regularizations on the output logits of T, S and G. This prevents our losses from being NaN when the logits grow too big. Specifically, we define $\mathcal{L}_{\mathtt{S}}$ as follows:

$$\mathcal{L}_{\mathtt{S}} \triangleq \lambda_0 \mathbb{E}_{z \sim p(z), x = \mathtt{G}(z)} \left[ \mathcal{L}_{\mathrm{KD}}(x) + \zeta_0 \max(|\mathtt{S}(x)| - \delta, 0) \right] +$$
$$\lambda_1 \mathbb{E}_{z' \sim p(z), x' = \tilde{\mathtt{G}}(z')} \left[ \mathcal{L}_{\mathrm{KD}}(x') + \zeta_0 \max(|\mathtt{S}(x')| - \delta, 0) \right] \tag{10}$$

where $\max(|\mathtt{S}(\cdot)| - \delta, 0)$ ensures that the output logit of S is between $[-\delta, \delta]$; $\zeta_0 \geq 0$ is a coefficient.

And we define $\mathcal{L}_{\mathtt{G,E}}$ as follows:

$$\mathcal{L}_{\mathtt{G,E}} \triangleq \mathbb{E}_{z \sim \mathcal{N}(0, \mathrm{I}), y \sim \mathrm{Cat}(C), u = \mathtt{G}_{\mathrm{lg}}(z+y), x = \sigma(u)} \left[ \vphantom{\sum} \right.$$
$$- \lambda_2 \mathcal{L}_{\mathrm{KD}}(x) + \lambda_3 \mathcal{L}_{\mathrm{NLL}}(x, y) + \lambda_4 \mathcal{L}_{\mathrm{NormReg}}(e_y)$$
$$\left. + \zeta_1 \max(|\mathtt{T}(x)| - \delta, 0) + \zeta_2 \max(|u| - \nu, 0) \right] + \lambda_5 \mathcal{L}_{\mathrm{BNmm}} \tag{11}$$

where $\mathtt{G}_{\mathrm{lg}}$ denotes the generator that produces logits instead of normalized images; $\sigma(\cdot)$ denotes the sigmoid function; $\max(|u| - \nu, 0)$ ensures that the output logit $u$ of $\mathtt{G}_{\mathrm{lg}}$ is between $[-\nu, \nu]$; $\max(|\mathtt{T}(x)| - \delta, 0)$ ensures that the output logit of T w.r.t. the synthetic sample $x$ is between $[-\delta, \delta]$; $\zeta_1, \zeta_2 \geq 0$ are coefficients.

We train MAD on multiple NVIDIA A100-SXM2-32GB and A100-SXM4-40GB GPUs. Due to the use of different teacher/student architectures, the use of GPUs with different numbers and types, and the share of computational resources, it is hard to compute exactly the training time of our method but roughly it took about 1-2 days, 3-4 days, and 4-6 days to train MAD on CIFAR10/100, TinyImageNet, and ImageNet/Places365/Food101, respectively.

| Dataset | Arch. | Student | | | | | Generator | | | | |
|---|---|---|---|---|---|---|---|---|---|---|---|
| | | $opt_{\text{S}}$ | $lr_{\text{S}}$ | $wd_{\text{S}}$ | $mo_{\text{S}}$ | $n_{\text{S}}$ | $opt_{\text{G}}$ | $lr_{\text{G}}$ | $wd_{\text{G}}$ | $mo_{\text{S}}$ | $n_{\text{G}}$ |
| CIFAR10/100 | ♡ | SGD | 1e-2 | 5e-4 | 0.9 | 30 | Adam | 1e-3 | 5e-4 | - | 3 |
| | ◇ | | | | | 60 | | | | | |
| TinyImageNet | ♡ | | | | | 90 | | | | | |
| ImageNet | ♣ | Adam | 1e-4 | 1e-4 | - | 150 | | 1e-4 | | | 20 |
| Places356 | | | | | | | | | | | 10 |
| Food101 | | | | | | | | | | | 5 |

Table 5: Settings of optimizers for student and generator w.r.t. different datasets and architectures. The teacher/student architecture settings are ResNet34/ResNet18 (♡), WRN40-2/WRN16-2 (◇), and AlexNet/AlexNet (♣). Meanings of abbreviations: $opt$: optimizer, $lr$: learning rate, $wd$: weight decay, $mo$: momentum, $n$: number of optimization steps.

| Dataset | Training settings | | | | | | | | | | | | |
|---|---|---|---|---|---|---|---|---|---|---|---|---|---|
| | $bs$ | $ld$ | $ldep$ | $ep$ | $spe$ | $d_z$ | $\alpha$ | $cg$ | $\gamma$ | $pg$ | $pgs$ | $\delta$ | $\nu$ |
| CIFAR10/100 | 256 | 0.1 | 100, 200 | 300 | 50 | 256 | 0.95 | No | - | No | - | | |
| TinyImageNet | | | | | | | | | - | | | | |
| ImageNet | 512 | - | - | 6000 | 1 | | | Yes | 1.1 | Yes | 200 | 20 | 20 |
| Places365 | | | | | | | | | 1.1 | | 50 | | |
| Food101 | | | | 4000 | | | | | 1.1 | | | | |

Table 6: Training settings of MAD w.r.t. different datasets. Meanings of abbreviations: $bs$: batch size, $ld$: learning rate decay, $ldep$: epochs at which learning rate are decayed, $ep$: total number of epochs, $spe$: steps per epoch, $d_z$: dimensionality of the noise $z$, $\alpha$: momentum for updating $\tilde{\text{G}}$, $cg$: G is class-conditional or not, $\gamma$: the scaling hyperparameter in Eq. 8, $pg$: Pretraining G or not, $pgs$: Number of steps for pretraining G, $\delta$: the bound in Eqs. 10,11, $\nu$: the bound in Eq. 11.

| Dataset | Student | | | Generator | | | | | |
|---|---|---|---|---|---|---|---|---|---|
| | $\lambda_0$ | $\lambda_1$ | $\zeta_0$ | $\lambda_2$ | $\lambda_3$ | $\lambda_4$ | $\lambda_5$ | $\zeta_1$ | $\zeta_2$ |
| CIFAR10/100 | 1.0 | 1.0 | 0.01 | 1.0 | 0.0 | 0.0 | 1.0 | 0.1 | 0.1 |
| TinyImageNet | | | | | | | | | |
| ImageNet | | | 0.1 | | | | | | |
| Places365 | | | 0.1 | | 0.1 | 0.1 | 0.0 | | |
| Food101 | | | 0.1 | | | | | | |

Table 7: Coefficients of the loss terms in $\mathcal{L}_{\text{S}}$ (Eq. 10) and $\mathcal{L}_{\text{G}}$ (Eq. 11).

## A.4  Generator Architectures

In Table 8, we show different architectures of the generator w.r.t. different image sizes.

**(a) CIFAR10/CIFAR100**

| Layer | Output size |
| --- | --- |
| Linear($d_z$, 4096) | 4096 |
| Reshape (16384, (256, 8, 8)) | 256×4×4 |
| ReLU() | 256×4×4 |
| BatchNorm2d(256, 0.1) | 256×4×4 |
| UpsamplingBilinear2d(2) | 256×8×8 |
| ConvBlockX(256, 128) | 128×8×8 |
| UpsamplingBilinear2d(2) | 128×16×16 |
| ConvBlockX(128, 64) | 64×16×16 |
| UpsamplingBilinear2d(2) | 64×32×32 |
| ConvBlockX(64, 32) | 32×32×32 |
| Conv2d(32, 3, 1, 0, 1) | 3×32×32 |

**(b) TinyImageNet**

| Layer | Output size |
| --- | --- |
| Linear($d_z$, 16384) | 16384 |
| Reshape (16384, (256, 8, 8)) | 256×8×8 |
| ReLU() | 256×8×8 |
| BatchNorm2d(256, 0.1) | 256×8×8 |
| UpsamplingBilinear2d(2) | 256×16×16 |
| ConvBlockX(256, 128) | 128×16×16 |
| UpsamplingBilinear2d(2) | 128×32×32 |
| ConvBlockX(128, 64) | 64×32×32 |
| UpsamplingBilinear2d(2) | 64×64×64 |
| ConvBlockX(64, 32) | 32×64×64 |
| Conv2d(32, 3, 1, 0, 1) | 3×64×64 |

**(c) ImageNet/Places365/Food101**

| Layer | Output size |
| --- | --- |
| Linear($d_z$, 8192) | 8192 |
| Reshape (8192, (512, 4, 4)) | 512×4×4 |
| ResNetBlockY(512, 512) | 512×4×4 |
| UpsamplingBilinear2d(2) | 512×8×8 |
| ResNetBlockY(512, 256) | 256×8×8 |
| UpsamplingBilinear2d(2) | 256×16×16 |
| ResNetBlockY(256, 128) | 128×16×16 |
| UpsamplingBilinear2d(2) | 128×32×32 |
| ResNetBlockY(128, 64) | 64×32×32 |
| UpsamplingBilinear2d(2) | 64×64×64 |
| ResNetBlockY(64, 32) | 32×64×64 |
| UpsamplingBilinear2d(2) | 32×128×128 |
| ResNetBlockY(32, 16) | 16×128×128 |
| UpsamplingBilinear2d(2) | 16×256×256 |
| ResNetBlockY(16, 16) | 16×256×256 |
| Conv2d(16, 3, 3, 1, 1) | 3×256×256 |

Table 8: Architectures of the generator w.r.t. different datasets. Details about ConvBlockX and ResNetBlockY are provided in Table 9.

| ConvBlockX($c_i$, $c_o$) |
| --- |
| Conv2d($c_i$, $c_o$, 3, 1, 1) |
| ReLU() |
| BatchNorm2d($c_o$, 0.1) |
| Conv2d($c_i$, $c_o$, 3, 1, 1) |
| ReLU() |
| BatchNorm2d($c_o$, 0.1) |

| ResNetBlockY($c_i$, $c_o$) | |
| --- | --- |
| Conv | Conv2d($c_i$, $c_o$, 3, 1, 1) |
|  | LeakyReLU(0.2) |
|  | BatchNorm2d($c_i$, 0.01) |
|  | Conv2d($c_o$, $c_o$, 3, 1, 1) |
|  | LeakyReLU(0.2) |
|  | BatchNorm2d($c_o$, 0.01) |
| Shortcut | Conv2d($c_i$, $c_o$, 1, 0, 1)  if $c_i \neq c_o$ 
 Identity()  otherwise |
| Output | $y = \text{Conv}(x) + \text{Shortcut}(x)$ |

(a)          (b)

Table 9: Architectures of ConvBlockX (a) and ResNetBlockY (b).

# B  Additional Experimental Results

## B.1  Results of Teacher

Table 10 reports the results of our teacher on all the benchmark datasets. On the small datasets, our teacher achieves very similar performance compared to the one in [14].

| | CIFAR10 | | CIFAR100 | | TinyIN | ImageNet | Places365 | Food101 |
| --- | --- | --- | --- | --- | --- | --- | --- | --- |
| | ResNet34 | WRN40-2 | ResNet34 | WRN40-2 | ResNet34 | AlexNet | AlexNet | AlexNet |
| Ours | 95.46 | 94.65 | 78.55 | 75.65 | 66.47 | 56.52 | 50.80 | 65.15 |
| In [14] | 95.70 | 94.87 | 78.05 | 75.83 | 66.44 | - | - | - |

Table 10: Classification accuracies of our teacher and of the one in [14] on different datasets.

## B.2  Results of DFKD-Mem with Different Memory Sizes

In Fig. 7, we show the classification results of DFKD-Mem with different memory sizes on CIFAR100 and ImageNet. On CIFAR100, DFKD-Mem achieves the best result at memory size = 2048 but still underperforms ABM and MAD. On ImageNet, the performance of DFKD-Mem is proportional to the memory size and is highest at memory size = 8192. This result, however, is still worse than that of MAD. Figs. 3a,3b display the average distillation loss (avg $\mathcal{L}_{KD}$) curves of DFKD-Mem w.r.t. different memory sizes. We see that increasing the memory size increases the avg $\mathcal{L}_{KD}$ on memory samples but does not affect the avg $\mathcal{L}_{KD}$ on samples from G. It is because the avg $\mathcal{L}_{KD}$ on memory samples is very small compared to the counterpart on samples from G.

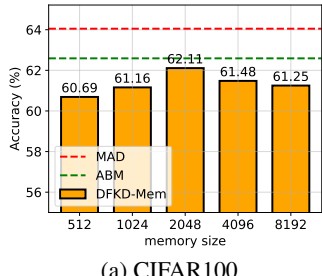
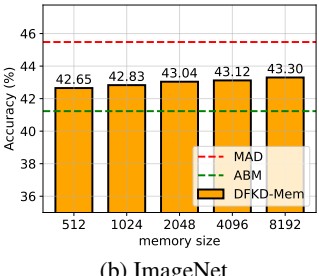

(a) CIFAR100                       (b) ImageNet

Figure 7: Classification accuracies of DFKD-Mem with different memory sizes and of MAD, ABM on CIFAR100 and ImageNet.

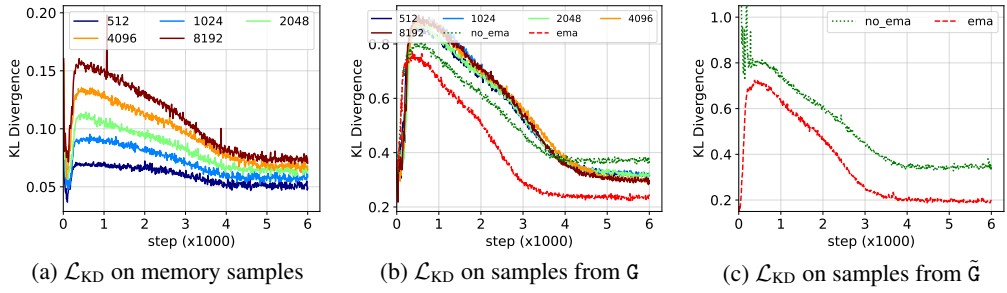

(a) $\mathcal{L}_{\mathrm{KD}}$ on memory samples     (b) $\mathcal{L}_{\mathrm{KD}}$ on samples from G     (c) $\mathcal{L}_{\mathrm{KD}}$ on samples from $\tilde{\mathrm{G}}$

Figure 8: Distillation loss curves on memory samples (a) and samples generated by G (b) and $\tilde{\mathrm{G}}$ (c). The numbers in the legends denote DFKD-Mem with the corresponding memory sizes. "no_ema" and "ema" denote ABM and MAD, respectively. The dataset is ImageNet.

## B.3    Empirical Analysis of Different Types of Generators

In Fig. 9, we show the results of MAD on ImageNet with three different types of generators which are unconditional ("uncond"), conditional-via-concatenation ("cat"), conditional-via-summation ("sum"). MAD with the "uncond" generator eventually collapses during training but not with the "cat" or the "sum" generators (Fig. 9a). This is because the "uncond" generator has learned to jump between different spurious solutions as visualized in Fig. 10. Among all types of generators, the "sum" generator enables stable training of our model and gives the best accuracy and crossentropy on $\mathcal{D}_{\mathrm{test}}$ (Figs. 9a,b). The "cat" generator only yields good results at $\lambda_3 = 0.3$ ($\lambda_3$ is the coefficient of $\mathcal{L}_{\mathrm{NLL}}$ in Eq. 6). The reason is that if $\lambda_3$ is too small (e.g., 0.1), $\mathcal{L}_{\mathrm{NLL}}$ will be high (Fig. 9g) and spurious solutions of G cannot be suppressed. G will jump between these solutions, leading to high variance when maximizing $\mathcal{L}_{\mathrm{KD}}$ (Fig. 9f). By contrast, if $\lambda_3$ is too big (e.g., 3.0, 10.0), G will be optimized towards predicting $y$ correctly (small $\mathcal{L}_{\mathrm{NLL}}$ as shown in Fig. 9g) rather than generating good adversarial samples for knowledge transfer from T to S (small $\mathcal{L}_{\mathrm{KD}}$ as shown in Fig. 9f). This causes S to achieve tiny $\mathcal{L}_{\mathrm{KD}}$ (Fig. 9e) and match T very well (Fig. 9d) on samples from G but generalizes poorly to unseen sample from $\mathcal{D}_{\mathrm{test}}$ (Fig. 9a). However, for any value of $\lambda_3$, MAD with the "cat" generator performs worse than the counterpart with the "sum" generator, and even worse than the counterpart with the "uncond" generator during early epochs of training (Fig. 9a). To explain this phenomenon, we first provide the formulas of the first layers of the three kinds of generators below as these generators are only different in the first layer:

$$\text{uncond: } h = Wz + b$$
$$\text{cat: } h = Wz + Ue_y + b$$
$$\text{sum: } h = Wz + We_y + b$$

where $W$, $U$, $b$ are trainable weights and bias. We hypothesize that due to the stochasticity of $z$ sampled from a *fixed* distribution, $W$ tends to be robust to changes. And since the "sum" generator uses $W$ to transform $e_y$, its output will not be affected much by the update of $e_y$. In other words, the

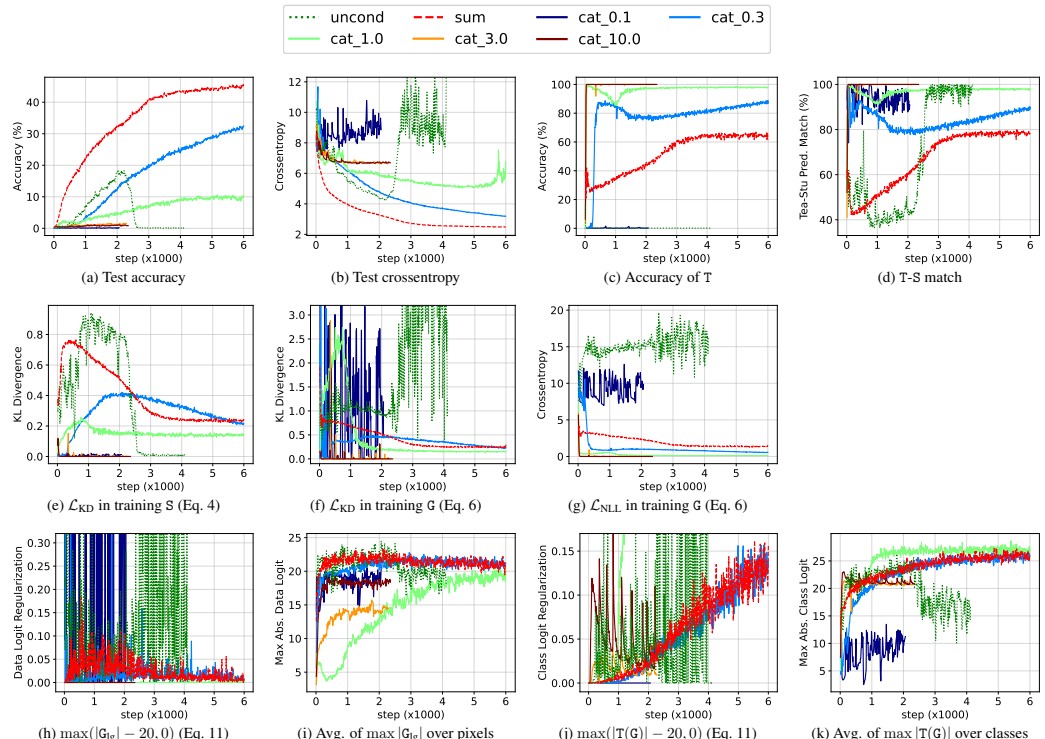

Figure 9: Various learning curves of MAD with different types of generators: unconditional ("uncond"), conditional-via-summation ("sum"), and conditional-via-concatenation ("cat"). For the "uncond" generator, $e_y$ is set to non-trainable zero vector and $\lambda_3$, $\lambda_4$ in Eq. 6 are set to 0. For "cat" generators, the number behind "cat" in the legend indicates the coefficient of $\mathcal{L}_{\text{NLL}}$ ($\lambda_3$) in Eq. 6. We tried different coefficients and found that $\lambda_3 = 0.3$ works best for the "cat" generator. Except for Test accuracy and Test crossentropy which are computed on samples $\mathcal{D}_{\text{test}}$, all other quantities are computed on synthetic samples from $\mathtt{G}$.

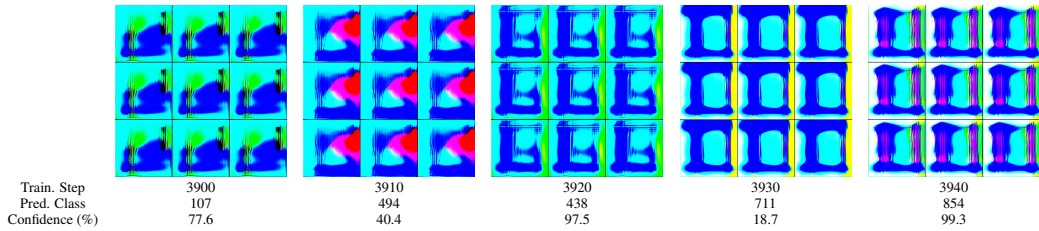

| Train. Step | 3900 | 3910 | 3920 | 3930 | 3940 |
|---|---|---|---|---|---|
| Pred. Class | 107 | 494 | 438 | 711 | 854 |
| Confidence (%) | 77.6 | 40.4 | 97.5 | 18.7 | 99.3 |

Figure 10: Visualization of generated samples from the unconditional generator whose has learning curves shown in Fig. 9. It is obvious that this generator jumps between different spurious solutions during training, which results in the collapse of the student in Fig. 9a.

noise in updating $e_y$ is absorbed into the stochasticity of $z$ via summation. The "cat" generator, on the other hand, uses a different weight matrix $U$ to transform $e_y$. Since the update of $U$ only depends on the current version of $e_y$ and vice versa, and $e_y$ can change arbitrarily, updating both $U$ and $e_y$ simultaneously in case of the "cat" generator can lead to unstable and nonoptimal[2] training. The "uncond" generator does not encounter any problem with $e_y$ like the "cat" generator so it can enable MAD to learn faster than the "cat" generator.

---

[2]During the backward pass at step $t$, $U_{t+1}$ is optimal for $e_{y,t}$ and $e_{y,t+1}$ is optimal for $U_t$. However, in the forward pass at step $t + 1$, $U_{t+1}$ is used for $e_{y,t+1}$ which leads to nonoptimality.

# C   Derivation of $\mathcal{L}_{\text{NormReg}}$ in Section 3.2

Recall that in our design of the class-conditional generator, we use $K$ *trainable* class embedding vectors $e_1, ..., e_K$ to represent $K$ classes in the training data. These embedding vectors can be regarded as the centers of $K$ Gaussian distributions (or clusters) $\mathcal{N}(e_k, \mathbf{I})$ ($k = 1, ..., K$) corresponding to $K$ classes and are optimized together with the generator $\mathsf{G}$ via Eq. 6. To prevent these embedding vectors from changing arbitrarily, we need to constraint their norms to be smaller than a threshold by minimizing the loss $\mathcal{L}_{\text{NormReg}}$ in Eq. 8. An important question is "What is a reasonable upper bound for the norm of each embedding vector $e_k$ ?".

Let $\xi$ denote the upper bound for the norm of $e_k$. By constraining $\|e_k\|_2$ to be smaller than $\xi$, we ensure that $e_k$ is inside a hyperball of radius $\xi$. Intuitively, we should choose $\xi$ so that the $K$ Gaussian clusters won't overlap each other. Note that in high dimensional space, we can generally treat each Gaussian cluster $\mathcal{N}(e_k, \mathbf{I})$ as a hypersphere of radius $\sqrt{d_e}$ centered at $e_k$ ($d_e = \dim(e_k)$). One simple way to allow these $K$ hyperspheres not to overlap each other when their centers are inside a hyperball of radius $\xi$ is to make sure that the total volume of $K$ hyperballs of radius $\sqrt{d_e}$ is smaller than the volume of the hyperball of radius $\xi$. Mathematically, it means:

$$K \times \mathcal{V}_{d_e}\left(\sqrt{d_e}\right) < \mathcal{V}_{d_e}(\xi)$$

$$\Leftrightarrow K \times \left(\sqrt{d_e}\right)^{d_e} \times \mathcal{V}_{d_e}(1) < \xi^{d_e} \times \mathcal{V}_{d_e}(1)$$

$$\Leftrightarrow K^{1/d_e}\sqrt{d_e} < \xi$$

where $\mathcal{V}_d(r)$ denotes the volume of a $d$-ball of radius $r$. When $d_e$ is large, $K^{1/d_e} \approx 1$ and can be ignored. Thus, we should choose $\xi$ to have the form $\gamma \times \sqrt{d_e}$ with $\gamma \geq 1$.