# OpenReview forum: "Momentum Adversarial Distillation: Handling Large Distribution Shifts in Data-Free Knowledge Distillation"
_NeurIPS.cc/2022/Conference — NeurIPS 2022 Accept_

### Official Review · Reviewer_Lniz · 2022-07-11

**Rating:** 5
**Confidence:** 4
**Soundness:** 3 good
**Presentation:** 3 good
**Contribution:** 2 fair

**Summary:**

In data-free knowledge distillation, a generator is used to synthesize data for training the student. This paper found that the distribution of synthetic data will change when the generator gets updated.  Thus, this paper proposes a new method called  Momentum Adversarial Distillation for data-free knowledge distillation. Experiments are done on six benchmark datasets to show the effectiveness of the proposed method.


**Questions:**

Using EMA to stabilize the model training is a common technique in existing works. This paper applies it to the generator of images for data-free knowledge distillation. Thus the novelty of the proposed method is limited. Another concern is from Figure5, the EMA generator fails to improve the quality of generated images, which makes me concerned about the cause of the improvement of student performance. How about adding EMA to the student?

**Ethics Review Area:**

["I don’t know"]

**Limitations:**

Yes.

**Strengths And Weaknesses:**

Strengths
1. The presentation of this work is clear and well-written.
2. The idea of the proposed method is straightforward.
3. I appreciate the sufficient experiments on different datasets and study on CIFAR100. They deliver the strong message that the proposed method is efficient.

Weakness
1. In figure5,  it is hard to conclude that the generated images by EMA Generator are better than the Generator. Could you use a metric to show how the difference for example the FID score?

2. In numerical experiments,  the results seems better than baselines. Results are mainly given the teacher and student are similar networks, how about the performance diverse architecture (resnet as teacher, alexnet as student)?

After rebuttal:
The reviewers have addressed my concerns. I insist my previous score after considering other reviewers' comments.

---

> ### Author Response · Authors · 2022-08-02
> **Our comments on the review of Reviewer Lniz (cont.)**
>
>
> ***[Question 1]*** *“Using EMA to stabilize the model training is a common technique in existing works. This paper applies it to the generator of images for data-free knowledge distillation. Thus the novelty of the proposed method is limited.”*: ***[Answer]*** Thank you for this insightful question. Your question is similar to Weakness 1 of Reviewer tq3D. We would like to provide our answer to your question below:
>
> We agree with Reviewer Linz that the idea of momentum update has been used in previous works. However, most of these works applied EMA update to the classifier/encoder to solve the semi-/self-supervised learning problem, with Mean Teacher [3] and MoCo [4] being two notable candidates. Compared to these works, our method is very different in terms of model design and motivation as specified below:
> * In terms of model design, in Mean Teacher/MoCo, the EMA update is applied to the classifier/encoder. Meanwhile, in our method, the EMA update is applied to the generator instead.
> * In terms of motivation, Mean Teacher/MoCo uses the EMA classifier/encoder to facilitate semi/self-supervised learning via forcing the outputs of the main classifier/encoder and the EMA classifier/encoder to be close. Meanwhile, we use the EMA generator to address the large distribution shift problem in adversarial DFKD, and we don't force the outputs of the generator and the EMA generator to be close but use samples from the EMA generator to train the student.
>
> We think the above differences constitute the novelty of our method.
>
> ***[Question 2]*** *“How about adding EMA to the student?”*: ***[Answer]*** Thank you for your interesting question. Your question is similar to Question 4 of Reviewer 5Fok. We would like to provide our answer to your question below:
>
> We tried to use the EMA student instead of the EMA generator on ImageNet and found that it just slightly improves the performance by about 0.4% which is very small compared to a 4% improvement when using the EMA generator. We hypothesize the reason is that using synthetic samples from the EMA generator helps the student recall old knowledge better than letting old knowledge decay slowly with the EMA student.
>
> **References**
> 1. Data Free Adversarial Distillation, Fang et al., arxiv 2019
> 2. Dreaming to Distill - Data-free Knowledge Transfer via Deep Inversion, Yin et al., CVPR 2020
> 3. Mean Teachers are better role models: Weight-averaged consistency targets improve semi-supervised deep learning results, Tarvainen and Valpola, NeurIPS 2018.
> 4. Momentum Contrast for Unsupervised Visual Representation Learning, He et al., CVPR 2020

---

> ### Author Response · Authors · 2022-08-02
> **Our comments on the review of Reviewer Lniz**
>
> Thank you for your insightful comments, we would like to address your concerns in detail below:
>
> ***[Weakness 1] + [Question 3]*** *“In Figure5, it is hard to conclude that the generated images by EMA Generator are better than the Generator. Could you use a metric to show how the difference for example the FID score?” + “Another concern is from Figure5, the EMA generator fails to improve the quality of generated images, which makes me concerned about the cause of the improvement of student performance.”*: ***[Answer]*** We would like to clarify that in Data Free Knowledge Distillation (DFKD), the performance of the student almost does not depend on the visual quality of generated samples as previous works empirically showed that the student can achieve very high accuracy of about 92-93% on CIFAR10 even when the generated images do not show any visually interpretable pattern of the original training data (Fig. 3 in [1], Fig. 3c in [2]). Moreover, the characteristic formula of adversarial DFKD (Eq. 2 in our paper) tends to force the generator $\mathtt{G}$ to generate samples that look different from the training data so that $\mathtt{G}$ could maximize the difference between the student and teacher. Thus, one cannot make any claim about the improvement of the student performance just by checking the visual quality of generated samples. In fact, a possible claim that we could make by visualizing synthetic samples in Figure 5 of our paper is that both the generator and the EMA generator can generate diverse samples and there is no sign of “mode collapse” (as presented in lines 260-263).
>
> We note that some methods like DeepInversion [2] could synthesize images that look quite like training samples. This is mainly attributed to the heuristic losses used by these methods (e.g., DeepDream loss, pseudo-cross-entropy loss, BatchNorm moment matching loss) rather than the adversarial loss between the generator and student. In our paper, we hardly used these heuristic losses (except for the BatchNorm moment matching loss in case of small datasets) but merely the adversarial loss. Therefore, it is reasonable that synthetic samples from our generator $\mathtt{G}$ do not look real. Similarly, synthetic samples from our EMA generator $\mathtt{\tilde{G}}$ also do not look real since $\tilde{\mathtt{G}}$ is the moving average of $\mathtt{G}$ over time.
>
> ***[Weakness 2]*** *“In numerical experiments, the results seem better than baselines. Results are mainly given the teacher and student are similar networks, how about the performance with diverse architectures (resnet as teacher, alexnet as student)?”*: ***[Answer]*** We thank Reviewer Lniz for this interesting question. In our paper, we did perform experiments with the teacher and the student being different networks. For example, we tested with the teacher/student being ResNet34/ResNet18 and WRN-40-2/WRN-16-2 on small datasets like CIFAR10, CIFAR100, and TinyImageNet. On large datasets like ImageNet, we set both the teacher and student to Alexnet to reduce the GPU memory and training time. We have run the setting that Reviewer Lniz suggested on ImageNet and have found that our method still significantly outperforms ABM - the baseline that does not use an EMA generator. Specifically, at training step 3000, our method achieves 8.32% accuracy while ABM only achieves 5.64%. This, again, validates the importance of the EMA generator in our method.
>
> In addition, thank to the suggestion of Reviewer Lniz, we observed an interesting phenomenon that the performance of the AlexNet student is much poorer when learning with the ResNet teacher than with the AlexNet teacher (8.32% compared to 40.66% for our method, and 5.64% compared to 34.52% for ABM at the training step 3000). After investigation, we have found some possible reasons for this listed below:
> * One possible reason is that the ResNet teacher is more complex than the AlexNet teacher so the AlexNet student has more difficulty mimicking the behavior of the ResNet teacher.
> * Another possible reason is that the training settings that we use for the AlexNet teacher and AlexNet student scenario are not suitable for the ResNet teacher and AlexNet student scenario. This could be true since, during training, we observed that the distillation loss of the student on synthetic samples increases instead of decreasing. It suggests that the student requires more optimization steps (larger $n_{\mathtt{S}}$). Other hyper-parameters may also need to be adjusted as well.
>
> We think this phenomenon should be investigated further. However, it does not change the fact that our method still performs better than the baseline that does not use an EMA generator.

---

### Official Review · Reviewer_tq3D · 2022-07-12

**Rating:** 4
**Confidence:** 4
**Soundness:** 2 fair
**Presentation:** 3 good
**Contribution:** 2 fair

**Summary:**

This paper shows that momentum adversarial distillation can help data-free knowledge distillation by solving the distribution shift problem. Extensive experiments are conducted on six datasets to demonstrate the effectiveness of the proposed method, including: CIFAR10, CIFAR100, Tiny-ImageNet, ImageNet, Places365 and Food101.

**Questions:**

-	Clarify the key contribution and novelty of the proposed method since the momentum updating method is not a new technique and has been widely studied.

-	Provide more explanations on how momentum update handles the distribution shift issue in the proposed framework.

-	The performance in the paper is not competitive and the comparison is not sufficient in the experiments.


**Limitations:**

This paper has discussed the limitations and negative social impacts.

**Strengths And Weaknesses:**

Strengths:

-	The proposed method is simple and seems effective, also easy to follow for other research.
-	The paper is generally well-written with sufficient experiments on six datasets.

Weaknesses:

-	The momentum updating method itself is not novel and has been used in many problems, such as semi-supervised learning [1], self-supervised learning [2], etc. I’m a little bit concerned about the novelty.

	[1] Tarvainen, Antti, and Harri Valpola. "Mean teachers are better role models: Weight-averaged consistency targets improve semi-supervised deep learning results." Advances in neural information processing systems 30 (2017).

	[2] He, Kaiming, Haoqi Fan, Yuxin Wu, Saining Xie, and Ross Girshick. "Momentum contrast for unsupervised visual representation learning." In Proceedings of the IEEE/CVF conference on computer vision and pattern recognition, pp. 9729-9738. 2020.

-	It seems the performance of the proposed method is not competitive if compared to other existing DFKD methods, as shown in Table 1. The comparison with other DFKD methods in Table 2 is also not sufficient.

-	There can be more analyses and visualizations to show how momentum update helps alleviate the distribution shift issue, not just the synthetic samples.

---

> ### Author Response · Authors · 2022-08-02
> **Our comments on the review of Reviewer tq3D**
>
> Thank you for your insightful comments, we would like to address your concerns in detail below:
>
> ***[Weakness 1] + [Question 1]*** *“The momentum updating method itself is not novel and has been used in many problems, such as semi-supervised learning, self-supervised learning, ....”* + *“Clarify the key contribution and novelty of the proposed method”*: ***[Answer]*** We thank Reviewer tq3D for this insightful comment. We agree with Reviewer tq3D that the idea of momentum update has been used in previous works, especially in the two famous models Mean Teacher and MoCo that Reviewer tq3D referred to. Our method is thus, inevitably inspired by these models. However, there are still some notable differences between our method and these models that we would like to highlight here:
> * In terms of model design, in Mean Teacher/MoCo, the EMA update is applied to the classifier/encoder. Meanwhile, in our method, the EMA update is applied to the generator instead.
> * In terms of motivation, Mean Teacher/MoCo uses the EMA classifier/encoder to facilitate semi/self-supervised learning via forcing the outputs of the main classifier/encoder and the EMA classifier/encoder to be close. Meanwhile, we use the EMA generator to address the large distribution shift problem in adversarial DFKD, and we don't force the outputs of the generator and the EMA generator to be close but use samples from the EMA generator to train the student.
>
> The above differences in motivation and model design between our method and Mean Teacher/MoCo constitute the novelty of our method. The significance of our method lies in its simplicity and adaptability to other problems (e.g., generalized continual learning, source-free domain adaptation, etc.) and data types. This was discussed in the Conclusion section (lines 295-299) of our paper and was appreciated by most reviewers.
>
> ***[Weakness 2] + [Question 3]*** *“It seems the performance of the proposed method is not competitive if compared to other existing DFKD methods, as shown in Table 1. The comparison with other DFKD methods in Table 2 is also not sufficient.”*: ***[Answer]*** In Table 1, it is clear that our method outperforms most existing DFKD methods. Our method only performs poorly than CMI and DFQ in some settings which we hypothesize due to their use of different losses and configurations for training. For example, CMI uses additional pseudo-cross-entropy loss [1] and inverse contrastive loss which are absent in our method; DFQ uses Variational Information Distillation (VID) loss [2] on intermediate features while our losses are only computed on the output logits. We discussed these differences in our paper from line 219 to line 222.
>
> Since our main target is to show that the EMA generator can help mitigate the large distribution shift problem rather than winning over current state-of-the-art methods, in Table 2, we only focused on the related baselines (ABM and DFKD-Mem) trained under the same settings as our method and didn't include other existing DFKD methods that do not support our main target. We explicitly stated that in our paper from line 223 to line 226. In addition, we note that all baselines in Table 1 have not been tested on large-scale datasets (ImageNet, Places365, Food101) so we have no clue about their desirable performances to put in Table 2.
>
> ***[Weakness 3] + [Question 2]*** *“Provide more analyses and visualizations on how momentum update helps alleviate the distribution shift issue, not just the synthetic samples”*: ***[Answer]*** We think in our paper, we conducted several experiments to show that EMA update helps alleviate the large distribution shift problem. We would like to list them below:
>
> * In Table 2, we showed that our method outperforms ABM - a baseline that does not use the EMA generator.
> * We conducted an ablation study on the weighting coefficient w.r.t. the EMA generator ($\lambda_{1}$ in Eq. 4) where we showed that reducing this coefficient leads to the decrease in the student's performance.
> * In Section 5.2.3, we empirically verified that the EMA generator's updates actually lead to smaller distribution shifts than the generator's updates.
>
> **References**
> 1. Data-free Learning of Student Networks, Chen et al., ICCV 2019
> 2. Variational Information Distillation for Knowledge Transfer, Ahn et al., CVPR 2019

---

### Official Review · Reviewer_qSj6 · 2022-07-12

**Rating:** 7
**Confidence:** 5
**Soundness:** 4 excellent
**Presentation:** 3 good
**Contribution:** 3 good

**Summary:**

This work focuses on Data-free Knowledge Distillation (DFKD). The main idea of existing DFKD is to use a generator to synthesize data for training the student. The authors propose Momentum Adversarial Distillation (MAD) to avoid forgetting the knowledge it acquired at the previous steps.

**Questions:**

Please see the weakness.

**Limitations:**

Yes

**Strengths And Weaknesses:**

Pros:

Many existing data-free knowledge distillation methods adopt an adversarial training paradigm, where the generator is adversarial trained for distillation, making the data distribution different from the earlier. This work focuses on the important yet understudied distribution shift problem in DFKD and propose a reasonable method to solve the problem.

The paper is overall well written and easy to follow. The experiments are well designed and the results well demonstrate the superiority of the proposed method.

Cons:

The experiments of this work have already addressed most my concerns about the paper. The only one remaining concern is that the difference or superiority of the proposed method from some highly related works should be clarified. For example, MosaicKD [1] also studies the domain shift problem in knowledge distillation, the authors should make the differences from the work clearer or make some comparisons if possible.

[1] Fang G, Bao Y, Song J, et al. Mosaicking to distill: Knowledge distillation from out-of-domain data[J]. Advances in Neural Information Processing Systems, 2021, 34: 11920-11932.

---

> ### Author Response · Authors · 2022-08-02
> **Our comments on the review of Reviewer qSj6**
>
> Thank you for your insightful comments, we would like to address your concerns in detail below:
>
> ***[Concern]*** *“The only one remaining concern is that the difference or superiority of the proposed method from some highly related works should be clarified. For example, MosaicKD [1] also studies the domain shift problem in knowledge distillation, the authors should make the differences from the work clearer or make some comparisons if possible.”*: ***[Answer]*** We thank Reviewer qSj6 for introducing this interesting work to us. We added a discussion about MosaicKD in the related work of our revised version. Here, we would like to summarize the difference between our method and MosaicKD:
> * *Difference in the problem setting*: Our method considers the setting in which no training data is provided to the student. Meanwhile, MosaicKD considers a less-extreme scenario in which some unlabeled, out-of-distribution (OOD) data is given to the student for distillation.
> * *Difference in the motivation*: Our method attempts to mitigate the large distribution shift problem in adversarial DFKD while MosaicKD tries to craft synthetic mosaic images using local patches extracted from the OOD data to facilitate knowledge transfer from the teacher to the student.
> * *Difference in the model design*: To achieve our goal, we propose to train the student with additional samples from the EMA generator. Meanwhile, MosaicKD uses a patch discriminator in addition to the student-teacher pair to guide the synthesis of synthetic mosaic images.

---

> > ### Comment · Reviewer_qSj6 · 2022-08-09
> > **Thank you for the reply**
> >
> > I have carefully read the responses and other reviewers' comments. My concerns have been properly addressed. I think this paper is a good complement to the field of data-free distillation, and have raised the score to 7.

---

> > > ### Author Response · Authors · 2022-08-09
> > > **Thank you for the response**
> > >
> > > Thank you for your response. We really appreciate your time and consideration.

---

### Official Review · Reviewer_5Fok · 2022-07-12

**Rating:** 6
**Confidence:** 4
**Soundness:** 2 fair
**Presentation:** 3 good
**Contribution:** 2 fair

**Summary:**

The paper tackles the problem of data free knowledge distillation, wherein access to a (teacher) model trained on some data is assumed, however the training data is not available. DFKD tries to train a student model using just the teacher model. The paper focuses on one method for DFKD which trains a generator network to generate synthetic examples for distilling the knowledge of the teacher. The main contribution of the paper is to use the generator along with an exponential moving average of the generator’s weights for producing samples. The paper claims that this improves training stability, since the data distribution that the student network sees does not change abruptly. This helps limit catastrophic forgetting of the student network on data from generators at previous iterations. The paper has a thorough empirical evaluation of the proposed method on multiple image benchmarks.

**Questions:**

1. Can the authors provide more evidence around how EMA helps with the catastrophic forgetting of the student? An ablation comparing the training and test accuracy curves with and without EMA could help

2. Can this trick of using an additional generator be plugged into other DFKD methods which rely on synthetic data? If so does it always help, or are other methods of making training more stable better?

3. How would the performance change if the train and test set have some distribution shift within themselves? In that case, would EMA help?

4. Can EMA also be applied on the student to stabilize the generator's training?

5. While EMA might mitigate catastrophic forgetting to some extent, it is also possible that there is a large shift within the EMA generator as well, leading to some amount of catastrophic forgetting. While this effect can be controlled by the alpha parameter of decay, an alternative could be to maintain an EMA of the EMA generator and so on. Do the authors have any insights around this?


**Limitations:**

Some of the limitations are listed in the weaknesses above. More analysis of the generators and their impact on the student would be appreciated.

**Strengths And Weaknesses:**

Strengths -
1. The paper proposes a simple trick to prevent catastrophic forgetting which does better than memory store techniques with lower overhead. The compute overhead is also lower than other DFKD methods.
2. The paper does extensive empirical evaluation of their method, along with a sensitivity analysis to various components of the approach.
3. The related works section is very well written.

Weaknesses -
1. The empirical evaluation is over a single run, so it is hard to ascertain the significance of the results. Adding error bars would definitely help here.
2. The claim of the student undergoing catastrophic forgetting is not sufficiently explored. While the experiments show that having a higher momentum (and weight for the ema generator’s images) is better to an extent, a class wise breakdown of where such models do better, and how this improvement is related to catastrophic forgetting could help.
3. The method is not interpretable. This is a general shortcoming of such adversarial methods, and should not be taken as a very specific shortcoming of this work.

---

> ### Author Response · Authors · 2022-08-02
> **Our comments on the review of Reviewer 5Fok (cont.)**
>
> ***[Question 3]*** *“How would the performance change if the train and test set have some distribution shift within themselves? In that case, would EMA help?”*: ***[Answer]*** We would like to clarify that in Data-Free Knowledge Distillation (DFKD), the student is not exposed to any training data and has to use synthetic data from the generator for training. The distribution shift of synthetic data is almost inevitable in adversarial DFKD since the generator has to continuously generate new samples to maximize the difference between the student and teacher. For evaluation, it is standard to use the original test set of the teacher which is usually drawn from the same distribution as that of the training set. One could consider out-of-distribution (OOD) test data but we think this setting is more suitable for the OOD generalization problem rather than the DFKD problem.
>
> ***[Question 4]*** *“Can EMA also be applied on the student to stabilize the generator's training?”*: ***[Answer]*** Thank you for this interesting question. We tried to use the EMA student instead of the EMA generator on ImageNet and found that it just slightly improves the performance by about 0.4% which is very small compared to a 4% improvement when using the EMA generator. We hypothesize the reason is that using synthetic samples from the EMA generator helps the student recall old knowledge better than letting old knowledge decay slowly with the EMA student.
>
> ***[Question 5]*** *“While EMA might mitigate catastrophic forgetting to some extent, it is also possible that there is a large shift within the EMA generator as well, leading to some amount of catastrophic forgetting. While this effect can be controlled by the alpha parameter of decay, an alternative could be to maintain an EMA of the EMA generator and so on. Do the authors have any insights around this?”*: ***[Answer]*** We think this is an interesting suggestion but could be costly in practice if we maintain multiple EMA generators. Therefore, we think it is better to stick with the simple yet efficient solution which is using a momentum decay alpha to control the update of the EMA generator.
>
> **References**
> 1. Data-Free Learning of Student Networks, Chen et al., ICCV 2019
> 2. Dreaming to Distill: Data-free Knowledge Transfer via DeepInversion, Yin et al., CVPR 2020
> 3. Data-Free Network Quantization With Adversarial Knowledge Distillation, Choi et al., CVPR Workshop 2020
> 4. Contrastive Model Inversion for Data-Free Knowledge Distillation, Fang et al., IJCAI 2021
> 5. Preventing Catastrophic Forgetting and Distribution Mismatch in Knowledge Distillation via Synthetic Data, Binici et al., WACV 2022
> 6. Generative adversarial network training is a continual learning problem, Liang et al., arxiv 2018

---

> ### Author Response · Authors · 2022-08-02
> **Our comment on the review of Reviewer 5Fok**
>
> Thank you for your insightful comments, we would like to address your concerns in detail below:
>
> ***[Weakness 1]*** *“The empirical evaluation is over a single run, so it is hard to ascertain the significance of the results. Adding error bars would definitely help here.”*:
> ***[Answer]*** We agree with this comment from Reviewer 5Fok. Due to the large number of runs we had to do (e.g., for hyper-parameter search and ablation study) and the long training time of each run (e.g., several days), it is costly to provide the standard deviation for our method and baselines. Therefore, in our paper, we only showed the result of one run. However, during our experiment, we tried to run some settings multiple times and observed that our method and related baselines usually have very small standard deviations among different runs. For example, the standard deviations of 3 different runs of our method on CIFAR100 and on ImageNet are just 0.18% and 0.32%, respectively. For ABM and ABM-Mem, the standard deviations are roughly the same. Meanwhile, our performance gains over ABM and DFKD-Mem are about 1.5-4% (Table 2) which makes the gain statistically significant. We note that in many related works (including those in Table 1) [1,2,3,4,5], the authors also did not show standard deviations, mainly because these values are small.
>
> ***[Weakness 2]*** *“The claim of the student undergoing catastrophic forgetting is not sufficiently explored. While the experiments show that having a higher momentum (and weight for the ema generator’s images) is better to an extent, a class wise breakdown of where such models do better, and how this improvement is related to catastrophic forgetting could help”*: ***[Answer]*** We would like to clarify the problem that the student suffers from catastrophic forgetting in adversarial DFKD has already been investigated in [5]. This problem also relates to the catastrophic forgetting problem of GANs which was analyzed in [6]. We discussed these papers in detail in our related work (starting from line 165). In our paper, we aim at providing a solution to this problem rather than justifying it. We hypothesize that the reason for the student's forgetting is due to the large distribution shift of synthetic samples from the generator $\mathtt{G}$. This is reasonable because the adversarial game between $\mathtt{S}$ and $\mathtt{G}$ forces $\mathtt{G}$ to continuously change its generated samples (on which $\mathtt{S}$ is trained) to maximize the difference between $\mathtt{S}$ and $\mathtt{T}$. Visualization of generated samples from an “uncond” generator in Fig 10 in our Appendix also provides some evidence to support this hypothesis. Therefore, we introduce an EMA generator to mitigate the large distribution shift induced by the generator, which in turn will mitigate the forgetting problem of the student.
>
> ***[Weakness 3]*** *“The method is not interpretable. This is a general shortcoming of such adversarial methods, and should not be taken as a very special shortcoming of this work.”*: ***[Answer]*** We thank Reviewer 5Fok for giving a fair judgment about the interpretability of our method. As in adversarial DFKD, the generator is forced to synthesize samples that maximize the difference between the student and the teacher, these synthetic samples may not look like the original training samples. Thus, the visual quality of synthetic samples cannot be used to interpret the results. In our paper, we tried our best to make our method as clear as possible via extensive experiments. For example, besides the main results in Tables 1 and 2, in Section 5.2.3, we compared the change in the updates of $\mathtt{G}$ and $\mathtt{\tilde{G}}$ to show that $\tilde{\mathtt{G}}$ can actually help mitigate the large distribution shift caused by $\mathtt{G}$.
>
> ***[Question 1]*** *“Can the authors provide more evidence around how EMA helps with the catastrophic forgetting of the student? An ablation comparing the training and test accuracy curves with and without EMA could help”*: ***[Answer]*** The comparison between our method and a variant that does not use an EMA generator was provided in Table 2 of our paper. In this table, ABM is the variant that does not use the EMA generator. A brief description of ABM was provided in lines 288 - 289 of our paper.
>
> ***[Question 2]*** *“Can this trick of using an additional generator be plugged into other DFKD methods which rely on synthetic data? If so does it always help, or are other methods of making training more stable better?”*: ***[Answer]*** We thank Reviewer 5Fok for this interesting question. We haven't tried to incorporate the EMA generator into other DFKD methods so we cannot say for sure but we think our idea of using an EMA generator can be favorably applied to existing DFKD methods especially those that continuously seek for new samples like in adversarial DFKD.

---

### Author Response · Authors · 2022-08-02
**Our general comment to all reviewers**

We are glad that the reviewers share common positive views about different aspects of our work which say that:
* **our proposed method** is: *“better than memory store techniques with lower overhead”* (Reviewer 5Fok), *“reasonable”* (Reviewer qSj6), *“simple and seems effective”*, *“easy to follow for other research”* (Reviewer tq3D), and *“straightforward”* (Reviewer Lniz).
* **our experiments** are: *“extensive”*, *“with a sensitivity analysis to various components of the approach”* (Reviewer 5Fok); *“well designed”*, *“well demonstrating the superiority of the proposed method”* (Reviewer qSj6), *“sufficient”* (Reviewer tq3D, Reviewer Lniz), and *“delivering the strong message that the proposed method is efficient”* (Reviewer Lniz).
* **our presentation** is: *“well-written”* (Reviewer 5Fok, Reviewer tq3D, Reviewer Lniz).

We also thank the reviewers for other insightful comments. We have revised our paper according to your suggestions. This makes the line numbers in the revised version different from those in the original submission. Therefore, we refer the reviewers to our original submission for the line numbers in our responses. We would like to address your concerns in detail below:

---

### Meta-Review · Area_Chair_RBGF · 2022-08-26

**Recommendation:** Accept
**Confidence:** Certain

**Metareview:**

This paper trains a generator to produce synthetic data for knowledge distillation from a teacher model, thus allowing distillation without the need of the original training data. The reviewers generally liked and had positive things to say about the method as well as the presentation, and the discussion was mostly around clarification and having better comparisons. This seemed to have satisfied the reviewers who responded to the rebuttals (not all of them did), and from my reading the authors did a good job at responding to concerns of already mostly positive reviews. The one negative review, which I felt was a little off-the-mark, I feel was addressed well by the rebuttals, but the reviewer dropped out afterwards and did not back up their ongoing criticisms.

I therefore recommend acceptance of this paper to NeurIPS.

Overall, discussion was rather limited, but this could be that the reviewers didn't have and serious concerns from the start and discussion was straightforward. I wish tq3D had contributed a little more as it would have been nice to arrive to a consensus.

**Award:**

No

---

### Decision · Program_Chairs · 2022-09-14

Accept